# Analysis of Biofilm Formation on the Surface of Organic Mung Bean Seeds, Sprouts and in the Germination Environment

**DOI:** 10.3390/foods10030542

**Published:** 2021-03-05

**Authors:** Marcin Kruk, Monika Trząskowska

**Affiliations:** 1Faculty of Human Nutrition, Warsaw University of Life Sciences—SGGW, Nowoursynowska 159c, 02-776 Warsaw, Poland; s187662@sggw.edu.pl; 2Food Hygiene and Quality Management, Department of Food Gastronomy and Food Hygiene, Institute of Human Nutrition Sciences, Warsaw University of Life Sciences—SGGW, Nowoursynowska 159c, 02-776 Warsaw, Poland

**Keywords:** biofilm, mung bean, organic food, probiotic, pathogen, sprouts

## Abstract

This study aimed to analyse the impact of sanitation methods on the formation of bacterial biofilms after disinfection and during the germination process of mung bean on seeds and in the germination environment. Moreover, the influence of *Lactobacillus plantarum* 299v on the growth of the tested pathogenic bacteria was evaluated. Three strains of *Salmonella* and *E. coli* were used for the study. The colony forming units (CFU), the crystal violet (CV), the LIVE/DEAD and the gram fluorescent staining, the light and the scanning electron microscopy (SEM) methods were used. The tested microorganisms survive in a small number. During germination after disinfection D2 (20 min H_2_O at 60 °C, then 15 min in a disinfecting mixture consisting of H_2_O, H_2_O_2_ and CH₃COOH), the biofilms grew most after day 2, but with the DP2 method (D2 + *L. plantarum* 299v during germination) after the fourth day. Depending on the method used, the second or fourth day could be a time for the introduction of an additional growth-limiting factor. Moreover, despite the use of seed disinfection, their germination environment could be favourable for the development of bacteria and, consequently, the formation of biofilms. The appropriate combination of seed disinfection methods and growth inhibition methods at the germination stage will lead to the complete elimination of the development of unwanted microflora and their biofilms.

## 1. Introduction

The consumption of unprocessed and organic food is gaining more popularity among consumers. There are many potential health benefits to eating such foods. Unprocessed fresh sprouted food supplies many nutrients, including proteins with high nutritional value, fibres and other carbohydrates, antioxidant compounds, vitamins and minerals [1]. Unfortunately, contamination of fresh produce by pathogens is a serious health issue in all countries. The main source of microbial contamination of bean seeds is the agricultural stage. The seeds are usually infected by a large number of microorganisms, including fungi, pseudomonads, enterobacteria, and lactic acid bacteria [2,3,4,5]. The bean seed microflora including mung bean may also be infected by pathogenic microorganisms, most often *Salmonella* and Shiga toxin-producing *Escherichia coli* (STEC) [6,7]. The production of mung bean sprouts must be carried out under controlled environmental conditions; the bean seeds are grown at 22–24 °C and high humidity in an aquatic environment for three to six days. Such production conditions and a large number of nutrients ensure easy growth of native seeds microflora [1,8,9]. From 1996 to July 2016, there were 46 reported outbreaks of foodborne illness in the United States associated with sprouts. These outbreaks accounted for 2474 illnesses, 187 hospitalizations, and three deaths [10]. Whereas in Germany and France in 2011, the highest number of food poisonings was recorded after the consumption of sprouts infected with *E. coli* STEC O104:H4. In total, this outbreak caused more than 3800 cases of illness and more than 50 deaths and involved several EU countries [11,12,13]. In addition, food poisoning is still reported after consuming raw sprouts. For example, in the USA, in 2020 alone, from January to July, almost 100 cases of poisoning with commercial sprouts were recorded [14]. Sanitation and disinfection of seeds before the cultivation of sprouts are methods that limit the development of microflora during the seed germination process. The most common method of seeds sanitation is soaking them in a water solution of chlorine because of its broad antimicrobial activity [15,16]. Nonetheless, chlorine compounds can be inactivated by organic substances that are present in food and may form various carcinogenic organochlorine compounds that threaten human health, therefore other methods of plant seeds sanitation are used [17,18]. In recent years, many methods of disinfection have been developed, consisting of one or more sanitation factors. The methods used are based on physical, chemical and biological factors that are neutral to human health. Physical methods are based on the use of hot water in various temperature ranges, high pressure, ultrasonic methods, gamma radiation, UV irradiation methods and even plasma activated water (PAW). Among the chemical methods for seeds sanitation, organic acids, hydrogen peroxide, ethyl alcohol and sodium chloride solutions are the most commonly used. Some authors also use biological methods of limiting the growth of pathogenic bacteria; such methods include the use of bacteriophages, bacteriocins and probiotic bacteria, mainly lactic acid bacteria. The combined methods (i.e., hurdle technology) bring the best results without significantly losing the quality of ready-to-eat sprouts [15,19,20,21,22,23,24,25,26]. Despite the effective sanitation treatment of the seeds, often they are re-inhabited by unwanted microflora during their cultivation. This is due to bacterial resistance to disinfection, uneven and rough surface of plant seeds, high content of nutrients in the seed environment, as well as existing of bacteria in the form of difficult-to-remove and complex biofilm structures [27,28]. Biofilms are a three-dimensional bacterial composition created as a result of the adhesion of microorganisms to the surface, which produce extracellular polymer substances (EPS). These polymers facilitate the development of bacteria in the biofilm, transport of nutrients and provide protection against external factors [29]. For this reason, biofilms are difficult to remove, and the bacteria present in them are difficult to inactivate [30]. The development methods of removing biofilms from food products such as plant seeds intended for sprouts while maintaining safety, high production efficiency and unchanged sensory characteristics are a challenge for the world of food science and technology. Prior to this, the knowledge about the dynamics of bacterial growth and the characteristics of changes in their number over time in the food production environment would be helpful in creating new solutions.

Therefore, this study aimed to analyse the impact of sanitation methods on the formation of bacterial biofilms on seeds after disinfection and during the germination process of mung bean and in the germination environment. Moreover, the influence of the biological sanitation factor on the growth of the tested pathogenic and saprophytic bacteria was researched.

## 2. Materials and Methods

A diagram of the experiment and the analyses performed is presented in Figure 1.

### 2.1. Mung Bean

Certified organic mung bean seeds (Bio Planet Inc., Leszno, Poland) were used for the research. The bean seeds were produced in China.

### 2.2. Bacterial Cultures

*Salmonella enteritidis* ATCC 13076, *Salmonella enteritidis* ATCC 29631, *Salmonella* Hofit IFM 2318, *Escherichia coli* ATCC 25922, *Escherichia coli* ATCC 10536 and *Escherichia coli* O157:H7 (isolated from FM0041-S10 Pathogens Control B, IFM, Ingleburn, Australia) were used as model bacteria. These strains were selected because they are the most common biological contamination of mung bean seeds and pose a threat to human health. *L. plantarum* 299v was isolated from the Sanprobi IBS dietary supplement (Sanprobi LLC., Szczecin, Poland) and served as a biological sanitation factor. All strains of *Salmonella* and *E. coli* were stored in brain heart infusion (BHI) broth (BioMaxima Inc., Lublin, Poland) with 20% (v/v) glycerol. *L. plantarum* 299v was stored in DE Man, Rogosa, Sharpe broth (MRS, Neogen Company, Heywood, UK) with 20% (v/v) glycerol. All of the strains were stocked at −80 °C. To prepare the inoculum, the bacterial strains were cultured on selective media as follows: Brilliant Green Agar (Oxoid, Basingstoke, UK) for *Salmonella*, MacConkey Agar with sorbitol, cefixime and tellurite (Neogen Company, Heywood, UK) for *E. coli* O157:H7, but Tryptone Bile X-Glucuronide agar (TBX, Bio-Rad LLC., Hercules, CA, USA) for other *E.coli* strains and MRS agar (Neogen Company, Heywood, UK) for *L. plantarum* 299v. The strains were cultivated at 37 °C per 24 h. A single colony of each strain without *L. plantarum* 299v was transferred to 10 mL of Buffered Peptone Water (BWP, Bio-Rad LLC., Hercules, CA, USA) and was incubated at 37 °C for 18 h to achieve a cell density of approximately 8 log CFU/mL. A single colony of *L. plantarum* 299v was transferred to 10 mL MRS broth and cultivated in the same conditions as other testes bacterial strains. *Salmonella* and *E.coli* cultures prepared in this way were used to cultivate the biofilm used in fluorescence analysis. For the bean seed inoculation, *L. plantarum* 299v was used as a biological sanitation factor; the bacteria cultures were centrifuged at 10,000× *g* rpm for 5 min. The obtained pellets were resuspended in 10 mL of sterile 0.85% saline.

### 2.3. Beans Inoculation

Mung bean seeds were disinfected to remove the native microflora before inoculating with the tested strains. The seeds were disinfected in sterile distilled hot water at 60 °C for 20 min according to Trząskowska et al. [31]. After disinfection, water was poured out, the seeds were cooled with sterile distilled water to 20 °C. The suspension of each strain was added to a separate batch of seeds in a ratio of 1.5:10. The bean was then mixed by vortexing for 5 min. The seeds were then placed on sterile plates and allowed to dry for 24 h. Drying took place in a biosafety chamber with constant airflow at 22 °C. During this time, the beans were stirred five times with a spatula. After this process, the initial bacterial count was 7 log CFU g^−1^.

### 2.4. Sanitizers and Treatments

Disinfecting treatments were carried out using two methods developed by Trząskowska et al. [31]. The mung bean seeds in the first method (D1) were treated for 10 min with sterile distilled hot water at 60 °C. After this time, the water was decanted and the seeds were poured over with the disinfecting mixture for another 10 min. In the second method (D2), the material was treated for 20 min with sterile distilled hot water at 60 °C. After this time, water was poured in and the seeds were poured over with the disinfecting mixture for 15 min. The disinfecting mixture consisted of 4% *v*/*v* H_2_O_2_ (30% *v*/*v* in H_2_O, Chempur, Piekary Śląskie, Poland), 0.2% *v*/*v* acetic acid (99.5% *v*/*v* in H_2_O, Chempur, Piekary Śląskie, Poland), and 95.8% *v*/*v* sterile distilled water. All components were mixed just before adding to the disinfected material. After sanitation, the mixture was poured and the material was washed 3 times with sterile distilled water to remove excess hydrogen peroxide from the surface. Test strips were used to detect residual hydrogen peroxide (Quantofix, Sigma-Aldrich Co., Poznań, Poland). At each described step, the material to liquid ratio was 1:2. There were 5 different seed variants for each strain: C (not disinfected with any method), D1 (disinfected with the first method), D2 (disinfected with the second method), DP1 (disinfected with the first method + *L. plantarum* 299v), DP2 (disinfected with the second method + *L. plantarum* 299v). The probiotic was added in the form of a cell suspension in 0.85% sterile saline; the number of cells was approximately 7 log/mL.

### 2.5. Analysis of the Biofilm

#### 2.5.1. Colony Formation Units

Bean seeds inoculated with the tested strains were cultivated in 50 mL Falcon tubes for 6 days at 22 °C. The weight ratio of the cultivated seed to the liquid in each case was 1:2. Every day, while growing the sprouts, the fluid was replaced with a new one. In samples C, D1 and D2, it was sterile distilled water. In samples DP1 and DP2, it was the aforementioned probiotic solution. The counting of bacteria was performed after seed disinfection, on the second, fourth and sixth day of sprouts cultivation. Additionally, in samples, DP1 and DP2, the enumerating of bacteria was done after 24 h of bean soaking. The initial number of each tested *E. coli* and *Salmonella* strain on bean seeds after inoculation and drying and before disinfection treatments was also determined. One gram of bean seeds was taken from each sample for analysis. Mung bean seeds were washed 3 times with sterile distilled water to get rid of planktonic bacteria. Then, 9 mL of sterile saline was added to the bean seeds and were vortexed for 5 min and 3000 rpm. The next step was to make decimal dilutions of the resulting bacterial suspension. Decimal dilutions were prepared in BPW. The diluted bacterial suspensions were spread on selective agars which were incubated at 37 °C for 24 ± 2 h for *S. enterica* and *E. coli* enumeration and 48 ± 2 h for *L. plantarum* enumeration. The analyses were performed in 3 replications.

#### 2.5.2. Crystal Violet Method

The crystal violet method used to classify the biofilm was adapted from [32,33,34]. The biofilm was grown in a 24-well microplate (NEST Biotechnology Co., Ltd., Wuxi, China) for 6 days at 22 °C. Two seeds of each analysed variant were placed in each well. The wells with C, D1 and D2 samples were filled with 500 µL of sterile distilled water. Meanwhile, the wells with samples DP1 and DP2 were filled with a probiotic suspension. Moreover, in this case, the ratio of seeds to liquid was 1:2. Every day during biofilm cultivation, the liquid was replaced with a new one. The biofilm was analysed each day of culture. The measurement of biofilm was as follows: after removing bean sprouts, the wells were washed 3 times with 1 mL of sterile distilled water to remove planktonic bacteria. The wells were dried for 45 min and the biofilm was fixed in 1 mL of 90% methanol (Chempur, Piekary Śląskie) for 30 min. Next, the wells were dried for 30 min, following 700 µL of crystal violet (Chempur, Piekary Śląskie, Poland) being poured into the wells and the biofilm being stained for 30 min. In the following step, crystal violet was removed and the plates were washed 6 times with distilled water and dried for 45 min. Thereafter, 1 mL of 33% acetic acid (Chempur, Piekary Śląskie, Poland) was poured into each well to extract the dye for 30 min. Finally, 150 µL of the acetic acid solution was collected from each well and poured into a 96-well plate (NEST Biotechnology Co., Ltd., Wuxi, China). The absorbance analysis was performed in a SpectraMax iD3 Multi-mode Microplate Reader (Molecular Devices LLC., San Jose, CA, USA) at a wavelength of 570 nm. A blank sample was prepared in the same way as the analysed biofilm. However, in the blind test, the seeds placed in the wells were not contaminated with bacteria but were disinfected twice by the second described method of sanitation. The biofilm formation capacity was assessed on the following calculation:Biofilm formation = *ODa* − *ODcn*
where *ODa* is the optical density of destained cells, *ODcn* is the optical density of negative control. The analyses were performed in 4 replications.

#### 2.5.3. Fluorescence Microplate LIVE/DEAD Method

The 96-well plate was filled with overnight cultures of *E. coli* and *Salmonella* strains. The 100 µL of microorganism culture was poured into each well and the plates were incubated for 24 h at 22 °C to form a pathogen’s biofilm. After this step, the bacterial suspension was removed and then each well was washed 3 times to remove planktonic bacteria; 200 µL of sterile distilled water was used to wash the wells. The biofilm in the wells was disinfected using the D1 and D2 method. C samples were used as a positive control. At each stage of sanitization, 200 µL of liquid was used. The biofilm was then detached from the wells with a sterile medical brush (Meringer Ltd., Kalisz, Poland). Each well was scraped for two minutes while vortexing at 500 rpm. Then, the detached bacterial cells were suspended in 100 µL of sterile microfiltered saline. Moreover, 100 µL of bacterial suspension was drawn for analysis and placed in a 96-well plate with clear bottom for fluorescent analysis (Brand GMBH + CO KG Wertheim, Germany). The procedure for staining and reading the fluorescence emission was performed according to the protocol LIVE/DEAD^®^ BacLight™ Bacterial Viability Kits containing SYTO 9 and Propidium iodide dyes (Molecular Probes Inc., Waltham, MA, USA). The method shows the ratio of the fluorescence emission in the 530 nm wavelength (emission 1; green) to the fluorescence emission in the 630 nm wavelength (emission 2; red). This part of the method is presented in the LIVE/DEAD protocol. The greater the value of the ratio obtained, the greater the number of viable cells in the bacterial suspension. The ratio green/red values have been correlated with the number of live cells (CFU). Correlation indices were determined using the values from the CFU analyses. The analyses were performed in 4 replications.

#### 2.5.4. Fluorescence Microplate Gram Method

The 96-well plate was filled with overnight cultures of *E. coli* and *Salmonella* strains. Overall, 100 µL of microorganism culture was poured into each well and the plates were incubated for 24 h at 22 °C to form a pathogen’s biofilm. After this step, the bacterial suspension was removed and then each well was washed 3 times to remove planktonic bacteria; 200 µL of sterile distilled water was used to wash the wells. The biofilm in the wells was disinfected using the D1 and D2 method. C samples were used as a positive control. At each stage of sanitization treatments, 200 µL of liquid was used. Mung bean seed extract was prepared as follows: mung bean seeds were combined with water in a weight ratio of 2:1; extraction was carried out by passive diffusion for 24 h at 22 °C; the obtained extract was microfiltered twice through a 0.2 µm membrane filter and filled into a sterile falcon tube. In this case, mung bean seed extract was used for the cultivation of biofilm to create conditions similar to the cultivation of sprouts environment. The *L. plantarum* 299v suspension in sterile saline was diluted decimal in mung bean extract. Then, 100 µL probiotic suspended was poured into all wells; the number of cells in 1 mL of probiotic liquid was approximately 7 log. For each tested strain, 3 variants of samples were obtained: DP1, DP2 and CP, i.e., samples not disinfected + *L. plantarum* 299v. The samples prepared in this way were incubated for 6 days at 22 °C. Every day during biofilm cultivation, the mung bean extract or probiotic was replaced with a new one solution. Fluorescence determinations were performed after the first, second, fourth and sixth day of incubation. Before analysis, the wells were emptied and washed 3 times with 200 µL of sterile distilled water to remove planktonic bacteria. Each well was scraped for two minutes while vortexing at 500 rpm. Then the detached bacterial cells were suspended in 100 µL of sterile microfiltered water. 100 µL of bacterial suspension was withdrawn for analysis and placed in a 96-well plate for fluorescent analysis (Brand GMBH + CO KG, Wertheim, Germany). The procedure for staining and reading the fluorescence emission was performed according to the protocol LIVE BacLight^TM^ Bacterial Gram Stain Kit containing SYTO 9 and hexidium iodide dyes (L-7005) (Molecular Probes Inc., Waltham, MA, USA). The method of interpreting the results was the same as for the LIVE/DEAD^®^ BacLight™ method. In this case, the ratio green/red indicates the number of gram-negative bacteria in the analysed sample. The analyses were performed in 4 replications.

#### 2.5.5. Light Microscopy

Microscopic analysis was performed using a microscope (Zeiss, Primo Star, Jena, Germany) with a camera (605100A 1/2″ 5 Mega Pixell Microscope Camera, Shanghai, China). The method described in research Oates et al. [35] was used for the analysis with modification. Bean sprouts were grown on the microscope slides under the same conditions as for the CFU method. The biofilm was analysed on microscope slides after removing the bean sprouts, washing off the planktonic bacteria and staining with the Gram method. Due to the direct contact of the bean seeds with the microscope slides, a bacterial biofilm was formed at the point of contact between the seeds and the microscope slides. This method was used to analyse biofilm formation after each method of sanitation for all strains of tested bacteria. The bacteria were stained with the standard Gram staining procedure. Microscopic images were taken after 6 days of biofilm cultivation. Bacteria were observed at 1000× magnification.

#### 2.5.6. Scanning Electron Microscopy (SEM)

Air-dried native, inoculated but untreated (C) and treated (D2 and DP2) mung bean was sputter-coated with gold. The coated seeds were examined by scanning electron microscopy (JSM—6390LV, JEOL Ltd., Tokyo, Japan).

The studies were performed in the Electron Microscopy Platform, Mossakowski Medical Research Centre Polish Academy of Sciences Warsaw, Poland.

### 2.6. Statistical Analyses

The statistical analysis of the results was performed using the Statistica 13.3 program (StatSoft, Kraków, Poland). The arithmetic means and standard deviation (SD) were calculated. For assessment, the data were normally distributed, and the Shapiro–Wilk test was carried out. A multi-factor analysis of variance ANOVA and Bonferroni post-hoc test was used to analyse the data. In the case of fluorescence methods, a correlation matrix with CFU results was performed to define the similarity of the results from both methods and Pearson’s correlation coefficient was calculated. The difference was considered statistically significant when *p* < 0.05. Error bars in figures and values after “±” in tables represent standard deviation.

## 3. Results

The impact of sanitation methods on the formation of bacterial biofilms on mung bean seeds and sprouts was measured by the colony forming units method. The results presented in Table 1 and Table 2 show the mean count of tested bacteria in the biofilm. The abundance of bacteria was significantly influenced by the sanitation method and the time of sprout cultivation (*p* < 0.05). These differences occurred with each tested strain. In the case of the *E. coli* ATCC 10536 strain, significant differences occurred between almost all tested samples. Every disinfection method used and the addition of a probiotic influenced the number of bacteria located in the biofilm during sprout cultivation. Nevertheless, on the last day of germination, the number of bacteria for all samples was at the level of 7 or 8 log CFU g^−1^. Regarding *E. coli* ATCC 25922, the number of bacteria increased with each day of incubation. The exceptions were samples treated with the D2 and DP2 methods, where, on the second day of cultivation, growth was significantly inhibited compared to the rest (*p* < 0.05). In addition, on the fourth day of incubation of the DP2 sample, biofilm development was still statistically lower than that of the other samples. The increase from the beginning to the 4th day of the sprouts’ incubation was, on average, 5.5 log CFU g^−1^. Thus, the D2 and DP2 methods had the greatest influence on the development of the *E. coli* ATCC 25922 biofilm. The addition of *L. plantarum* 299v delayed the expansion of the undesirable biofilm of this *E.coli* strain in the most significant way. Similarly, in the case of *E.coli* o157:H7, D2 and DP2 were also the most effective sanitation methods. An interesting phenomenon was observed during the incubation of the DP2 sample. On the fourth day of cultivation probe DP2, the number of bacteria decreased by almost 3 log CFU g^−1^ compared to the number of bacteria determined the day before, which was 6.13 (SD = 0.05) log CFU g^−1^ (*p* < 0.05). Despite these variations, the number of bacteria on the last day of sprouting increased to an average of 6.8 log CFU g^−1^. Overall, the disinfection methods DP2 had the greatest effect on the growth of *E. coli* strains to the 4th day of biofilm formation compared to the results of the D2 method, but on the last day of germination, bacterial count did not differ significantly from the non-disinfected control.

The *S. enteritidis* ATCC 13076 and *S.* Hofit IFM 2318 strains were characterized by a similar ability to form biofilms after the tested disinfection procedure. Significant differences in the number of bacteria compared to the control sample occurred just after disinfection. In the case of the 13076 strain, the D2 and DP1 methods considerably limited the process of biofilm development also on the second day of sprout cultivation (*p* < 0.05). If it was the 2318 strain, then only the DP2 method significantly changed the number of bacteria in the biofilm compared to other samples on the second day of incubation. Over the following days of sprout cultivation, the bacterial number in the biofilm increased to 7–8 log CFU g^−1^ and there were no significant differences between the samples. The biofilms of the *S. enteritidis* ATCC 29631, created on the mung seed cover, were uniquely susceptible to the tested sanitation methods. All the disinfection methods used resulted in a significant decrease in the number of *Salmonella* cells and effectively prevented the re-development of bacterial structures on and in sprouts. Only concerning the D1 method, after the entire period of sprout cultivation, the count of *S. enteritidis* ATCC 29631 was 2.91 (SD = 0.37) log CFU g^−1^. Meanwhile, the number of bacteria in the control sample after the total incubation period was 8.56 (SD = 0.36) log CFU g^−1^. In the case of the DP1 method, the growth of pathogenic microflora took place during the second and fourth day of cultivation. On the sixth day, a significant decrease in the number of these bacteria from 3.67 (SD = 0.17) log CFU g^−1^ to 0.91 log CFU g^−1^ were reported. What is more, a significant effect of the addition of the probiotic bacteria on the number of pathogenic bacteria was observed (*p* < 0.05). The most effective disinfection methods regarding this strain were D2 and DP2. No significant growth of the number of *S. enteritidis* ATCC 29631 was observed during the entire period of sprout cultivation after used disinfection methods. Summarizing, beside the influence of the disinfection methods and the time of sprout cultivation on biofilm formation, the susceptibility of the tested *Salmonella* strains plays a role (*p* > 0.05).

To some extent, the above results are supported by the scanning electron microscopy (SEM) images of the disinfected mung bean surface. The whole seed, hilum, and coat of air-dried mung bean examined by SEM are presented in Figure 2 and Figure 3. The micrographs of native mung bean show the openings in the hilum and the cracks on the coat. However, we do not observe any significant presence of microorganisms (Figure 2). After inoculation, pathogens colonized the surface of the cover and adhered to the hilum structures (Figure 3(1A,2A)). The applied, multi-stage disinfection method (D2 and DP2) limited the number of visible microorganisms and negatively influenced the cell structure (Figure 3(1B,1C,2B,2C)). The addition of the next stage of disinfection, i.e., the biological agent (*L. plantarum* 299v), resulted in the colonization of the surface of sanitized beans by this bacterium as cells in very good condition (Figure 3(1C,2C)). Unfortunately, the uneven surface makes it possible for individual cells to survive and develop during germination.

For the analysis of the biofilm formation in the germination environment, the crystal violet method was used, among others. Thanks to this, the microbiological activity in the germination environment, i.e., the surfaces and materials used for germination, was demonstrated. The changes in biofilm development measured by the crystal violet (CV) method are shown in Figure 4.

The biofilms developed by *E. coli* strains were similar and significantly influenced individually by the sanitation method and the time of sample incubation, and the interaction between these factors was observed (*p <* 0.05). There were no significant differences among the samples treated with probiotic bacteria (DP1; DP2) (*p* > 0.05) and in between the samples without *L. plantarum* 299v (C; D1; D2) (*p* > 0.05). However, samples C, D1 and D2 differed significantly from the samples with probiotic (DP1, DP2) (*p* < 0.05). These differences occurred within all tested strains of *E.coli*. However, with strain *E. coli* ATCC 10536 (Figure 4A), there was a substantial difference between sample D1 on the 5th day of incubation for samples C and D2 (*p* < 0.05) and was not significantly different from the DP1 sample (*p* > 0.05). To sum up, the amount of biofilm formation researched by CV method was mainly influenced by the addition of a probiotic culture to the germination stage. In the case of *Salmonella*, the most significant increase in bacterial biofilm was after the third day of incubation, but this observation does not apply to samples DP1 and DP2.

Over the following days of incubation, there was the greatest increase in the amount of biofilm. The development of the *S. enteritidis* ATCC 13076 (Figure 4D) biofilm followed similar dynamics in samples C, D1 and D2. Differences in the amount of biofilm formed for these strains were not significant for the individual days of incubation (*p* > 0.05). This means that the increase in the amount of biofilm between the samples occurred in similar dynamics.

An important aspect is that the final amount of biofilm for samples C, D1 and D2 was not significantly different (*p* > 0.05) from the samples with *L. plantarum* 299v (DP1; DP2). The amount of biofilm formed by *S. enteritidis* ATCC 29631 (Figure 4E) within samples D1, D2 increased significantly after the third day of incubation (*p* < 0.05). An interesting phenomenon was observed between sample C and samples D1; D2, while in sample C, the growth of biofilm was not as dynamic as in compared probes and reached a value similar to the disinfected samples only on the last day of incubation. The amount of biofilm on the fourth, fifth and sixth days’ incubation samples D1 and D2 was similar to the biofilm amount by samples DP1 and DP2 (*p* > 0.05). In the case of *S.* Hofit 2318 (Figure 4F), significant differences in the amount of biofilm between samples with probiotics occurred mainly until the third day of incubation (*p* < 0.05). Until then, the DP2 and DP1 samples were significantly different from the C, D1 and D2 samples (*p* < 0.05). To sum up, the applied disinfection methods had a significant effect on the slowing down of biofilm formation up to the 3rd day of incubation of the samples (*p* < 0.05). After this period, there was intensive development of bacterial biofilms for C, D1 and D2 samples. In the case of probes with probiotics (DP1; DP2), the amount of biofilm remained relatively similar throughout the incubation period of the samples, with the trend of increasing its number (*p* > 0.05).

Another attempt to analyse the biofilm formation in the germination environment was harvesting them on microscopic slides, Gram staining and visualising in the light microscope. The images obtained as a result of the light microscopy analysis of the Gram-stained biofilm structure are presented in Figure 5. The red and pink coloured cells are Gram-negative bacteria; the blue and navy blue structures are Gram-positive bacteria. An important element of this analysis is the imaging of biofilms formed simultaneously by probiotic and pathogenic bacteria or saprophytic bacteria. Nevertheless, it was noticed that the biofilms formed by these two types of bacteria have a lower amount of pathogenic and saprophytic bacteria compared to samples without *L. plantarum* 299v. The only exception in the case of no biofilm formation during the sprout cultivation period was sample D2 of strain *S. enteritidis* ATCC 29631. In this sample, there were no bacterial biofilms, and only single bacterial cells adhered to the analysed material (Figure 5b). Besides, the results show that despite the use of mung bean seed disinfection methods, their germination environment was favourable for the development of bacteria and, consequently, the formation of biofilms.

When analysing the phenomenon of biofilm and the influence of disinfection on its formation, an important issue is to what extent it causes cell death. To evaluate this problem, the fluorescence microplate LIVE/DEAD method was applied. The results of the fluorescence assay were presented in juxtaposition with the CFU outcome to compare the data from both methods of analysis (Figure 6). The lowest obtained correlation coefficient for the matrix of results from the two methods was *r* = 0.88 and it concerned the relationship between the results for the *S.* Hofit IFM 2318 strain. The correlation coefficients for the remaining strains were *r* = 0.96 on average. The results are almost linearly related to each other. One-dimensional significance tests showed that the disinfection method had an effect (*p <* 0.05) on reducing the number of live bacteria in the samples. The data obtained from the post hoc analysis indicated that there were statistical differences (*p <* 0.05) in the count of viable bacteria between the non-disinfected samples and the disinfected samples for all tested bacterial strains. However, significant differences (*p <* 0.05) in the number of viable bacteria compared to the two applied disinfection methods occurred only in the case of *S. enteritadis* ATCC 13076, *E. coli* ATCC 10536 and *E. coli* ATCC 25922. For the remaining strains, there were no statistical differences in the number of viable bacteria between the D1 and D2 methods. Despite the lack of statistical differences for the *E. coli* O157:H7, *S.* Hofit IFM 2318 and *S. enteritadis* ATCC 29631, a lower number of viable bacteria was always observed after used the more intensive disinfection method (D2). Comparing to the CFU results, the samples where no CFU was detected, viable bacteria were detected based on the results of fluorescence staining.

Meanwhile, the fluorescence microplate Gram method was helpful to analyse the interaction between tested pathogens (Gram-) and probiotic bacteria (*L. plantarum* 299v; Gram +). When using the sanitation method with probiotic, the amount of gram-negative bacteria in biofilms was significantly influenced, as well as by the time of biofilm cultivation, (*p* < 0.05) (Table 3). For all tested strains, a lower fluorescence emitted by gram-negative bacteria was observed in the samples DP1; DP2 compared to the non-disinfected control (CP) on the first day of incubation (*p* < 0.05). On the second and fourth day of incubation, the number of bacteria in the disinfected samples increased. However, these differences were not significant (*p* > 0.05). It is related to the significant influence of the used sanitation method combined with the addition of a probiotic on limiting the development of the biofilm. The count of bacteria on the last day of incubation in samples DP1 and DP2 increased significantly compared to the data obtained from the previous days of incubation (*p* < 0.05). Furthermore, the Gram-negative bacteria in the samples from the last day approached that obtained in the CP, but was statistically lower regarding the amount of formed biofilm (*p* < 0.05). Only for *E. coli* O157:H7 and *E. coli* ATCC 25922, this relationship was not observed. It proves the significant influence of the DP disinfection method on limiting the development of undesirable microorganisms’ biofilm. Additionally, after applying the correlation to compare the results of this method and the CFU method, the average correlation coefficient for all samples was *r* = 0.50. This shows that the obtained results are consistent. To sum up, the disinfection methods with probiotic, had a significant (*p* < 0.05) impact on reducing the intensity of pathogenic biofilm development, although it was not strong enough to completely prevent its formation.

## 4. Discussion

The studies of the development of microflora after the seeds’ disinfection intended for sprouts production were conducted by many authors. The results of these research, despite the usage of various methods of disinfection, identified undesirable microflora grown during germination. The growth dynamic of bacteria presented in this experiment during germination was similar to the results of Zhang et al. [19] and Fransisca et al. [36]. However, they counted all bacteria in the sprouts. The similar values may indicate that most of the microorganisms contaminating food sprouts are found mainly in biofilms or assume an adhesive form. It was made visible thanks to microscopic methods, where the bacteria are present in clusters but also as a single cell (Figure 5). Interestingly, other authors who studied biofilms formed on food sprouts obtained results similar to those presented in this article. *Salmonella enteritidis* ATCC 29631 did not form a biofilm at all or with much fewer cells after disinfection with different methods. Li and Gänzle [37], He et al. [38], and Ma et al. [39] found that the resistance of bacteria to unfavourable factors depends not only on the species or type of bacteria but also on the strain. After decontamination, some bacteria of this strain have lost the ability to form a biofilm on the surface of the sprouts. However, the analysis of the biofilm in the germination environment shows more growth dynamics previously mentioned *Salmonella*, i.e., the formation of biofilms on polystyrene plates was observed. This variance of growth depending on the surface is reported in the literature [40,41]. Moreover, the other types of surface factors influencing the biofilm formation were: temperature, the physiological and metabolic state of microorganisms, and other external and internal factors [41]. The results of the CV analysis indicated that all tested strains were more or less capable of forming biofilms. The volume of biofilm formed varied between *E. coli* and *Salmonella*. Han et al. [28] and Kim et al. [42] obtained similar results. In these studies, less *E. coli* biofilm and more *Salmonella* biofilm were obtained. However, the final growth indicators were different due to the use of a different growth medium.

An important aspect of this analysis is comparing the CV results with the CFU outcome. The amount of biofilm obtained does not always indicate the proportional number of metabolizing bacteria in it. Based on the data obtained from own research and data available in the literature, it can be concluded that some bacteria are capable of producing more extracellular metabolites, which influence the amount of biofilm structures formed [43,44,45,46]. Even though *Salmonella* produced more biofilm than *E.coli*, the number of bacteria living in the biofilm was similar. This dependence can also be seen in microscopic images where the bacteria of the genus *Salmonella* form thicker structures (Figure 5).

Another aspect of the research is the introduction of *L. plantarum* 299v as a biological agent to limit the development of undesirable bacteria, especially in organic food processing. Maintaining microbiological quality in organic sprout production is a particular challenge due to the inability to use the most effective chlorine compounds for disinfection [47]. Enrichment with *L. plantarum* 299v had a positive effect on the nutritional value of legume sprouts [48]. Additionally, the genus of *L. plantarum* including 299v strain posed antimicrobial properties, including mesophilic bacterial counts in legume sprouts and other food products [21,49]. Based on the obtained results, no clear influence of this factor was found. In the case of the CFU assay, the final number of pathogenic bacteria was not significantly influenced by the use of this probiotic bacteria. However, the inactivating effect of *L. plantarum* 299v on pathogenic bacteria growth during the first 3 days of sample incubation was revealed and deserves underlining. This is in the line with the findings of Świeca et al. [20] and Rossi et al. [50], who demonstrated clear antimicrobial properties of *L. plantarum* in sprouted food products. Meanwhile, other authors using probiotic bacteria to limit the growth of pathogens on raw vegetables and fruits have found their ambiguous impact on the number of undesirable bacteria [42,51].

The next substantial aspect of the study was the survival of bacteria after disinfection methods. Numerous authors developed disinfection methods that resulted in microorganisms’ reduction at various levels. Some authors achieved an almost complete inactivation of bacteria after treatment [31,52]. However, despite the application of very effective disinfection methods, microflora and biofilms development during the seed germination could happen [19,31]. The most intensive development of biofilm took place after the second day of germination. The CFU, CV, and Gram fluorescence results showed that bacterial biofilms grew most after day two. It means that the second day of germination is the critical moment after which the safety of the sprouts decreased to the greatest extent. However, samples disinfected with the D2 and DP2 methods showed a better microbiological quality concerning the critical day, i.e., until the 4th day. One can assume that depending on the method used, the 2nd or 4th day could be a time for the introduction of an additional growth limiting or inactivating factor. In the studies conducted by Trząskowska et al. [31] and Warriner et al. [53] after the application of disinfection, only a few bacterial cells survived, which was enough for the development of a large number of bacteria after the incubation period of sprouts. This phenomenon is reflected in the presented research. As in other studies [54,55], obtained SEM micrographs demonstrate the uneven and cracked surface of mung bean cover, which pose favourable conditions for the survival of disinfection and the CFU detection below method’s limit (<10 CFU g^−1^) (Figure 2). For this reason, the LIVE/DEAD method gave more information about the viability of bacteria. The comparison of the results from the CFU and LIVE/DEAD methods shows that even in cases where the results of the CFU analysis were below the detection limit after disinfection, the LIVE/DEAD analysis data indicated viable bacteria in these samples (Figure 6). This proves the survival of a minuscule number of bacteria, which then during germination can grow, settle the sprouting environment and create biofilms. The seeds are covered with native biofilm settled on a rough and cracked surface, which is why removing or inactivating all microflora by disinfection becomes so difficult [28,56,57]. Some authors presented similar data obtained with the fluorescence technic. However, in the experiments, mainly microscopic methods were used, which indicated the presence of bacteria after sanitation of the test materials, but did not concern biofilms or their formation [53,58].

## 5. Conclusions

The formation of biofilms is a very complex and often rapid process. Moreover, these structures are difficult to remove. Despite the use of mung bean seed disinfection methods, their germination environment could be favourable for the development of bacteria and, consequently, the formation of biofilms. Combined disinfection methods before the germination process and the addition of probiotic bacteria for the cultivation of sprouts provide a great chance to limit the growth of pathogenic bacteria. This is especially important in organic food production where limited substances and methods are allowed. The appropriate combination of seed disinfection methods and growth inhibition factors at the germination stage will let to the complete elimination of the development of unwanted microflora and their biofilms. Depending on the method used, the 2nd or 4th day could be a time for the introduction of an additional growth-limiting factor. The results of our research indicate the imperfection of single research methods in the analysis of surviving bacteria. The multi-method approach detects the presence of cells below the detection limit of other methods, e.g., in the LIVE/DEAD method, the characteristic fluorescence of living cells was detected, while the CFU method did not detect viable microflora within the same sample. What is more, the place of the sprout germination process should be under strict control because microorganisms may attach to surfaces and grow well. This phenomenon may be a source of cross-contamination in the future.

However, the limitation of the experiment is the model system used. It is advisable to repeat the tests under the industrial conditions of sprout germination. The challenge for scientists and technologists continues to be the development of disinfecting methods, which will be used at the stage of sprout growth. At this point is the greatest risk of developing pathogenic microflora that threatens human health and life.

The obtained results allowed one to deepen the knowledge on the formation of these structures after sanitising and during the germination of mung bean seeds, along with proposals for critical points in sprout production for consideration.

## Figures and Tables

**Figure 1 foods-10-00542-f001:**
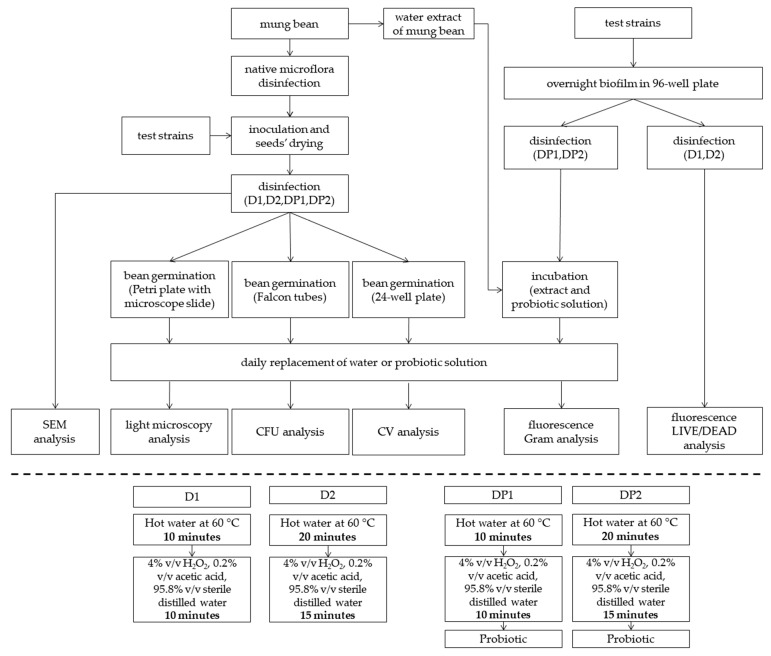
Schematic diagram of the experiment and disinfection methods (D1, D2, DP1, DP2); probiotic—*L. plantarum* 299v. D1—disinfected with the 1st method; D2—disinfected with the 2nd method; DP1—disinfected with the 1st method + *L. plantarum* 299v; DP2—disinfected with the 2nd method + *L. plantarum* 299v; SEM—Scanning Electron Microscopy; CFU—Colony Formation Units; CV—Crystal Violet method.

**Figure 2 foods-10-00542-f002:**
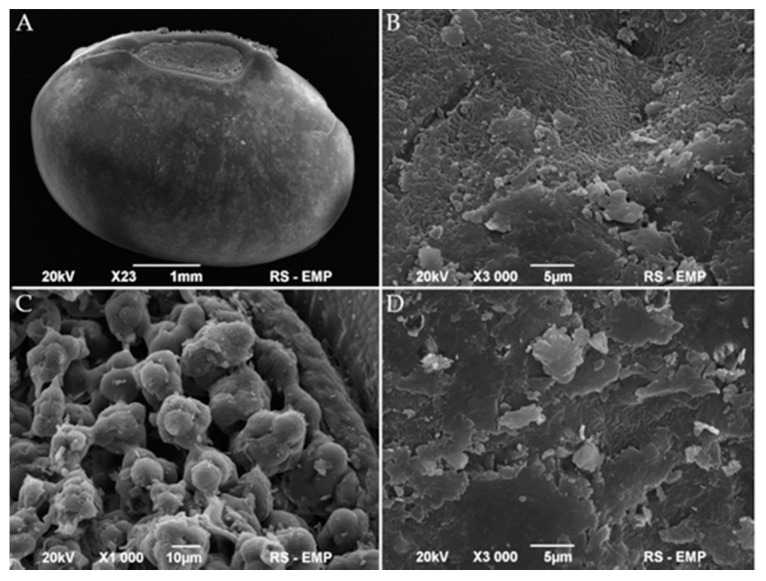
Scanning electron micrographs of whole native bean (**A**), hilum (**C**), and coat (**B**,**D**) of mung bean.

**Figure 3 foods-10-00542-f003:**
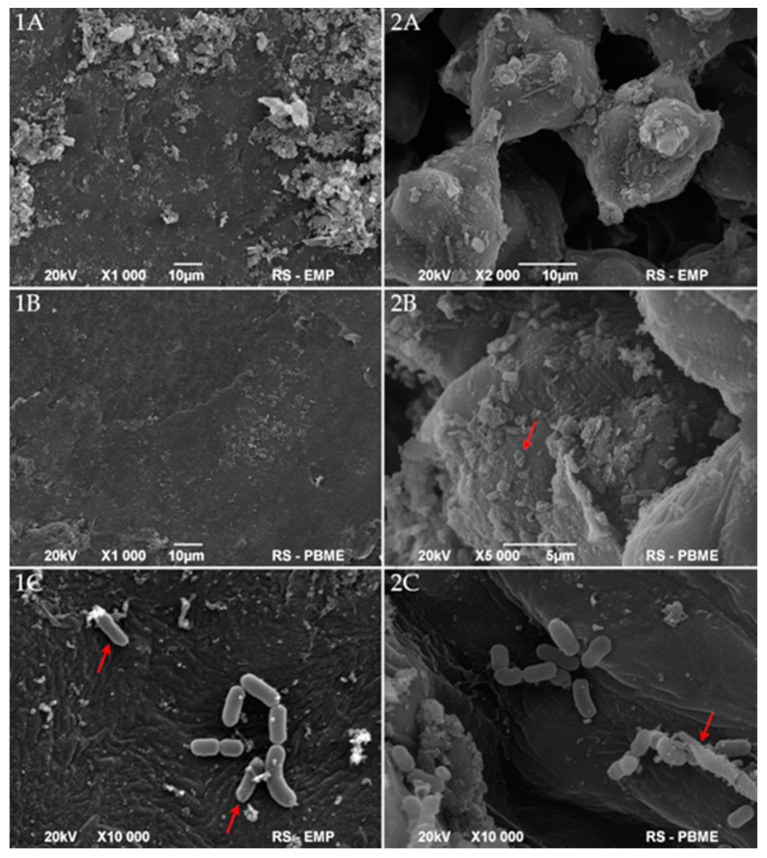
Scanning electron micrographs of coats (**1**) and hila (**2**) of mung bean inoculated with pathogens (**A**(**1**,**2**)) and after then disinfected with D2 method (**B**(**1**,**2**)) or DP2 method (**C**(**1**,**2**)); arrows indicate destroyed (**2**(**B**,**C**)) or damaged (**1C**) cells.

**Figure 4 foods-10-00542-f004:**
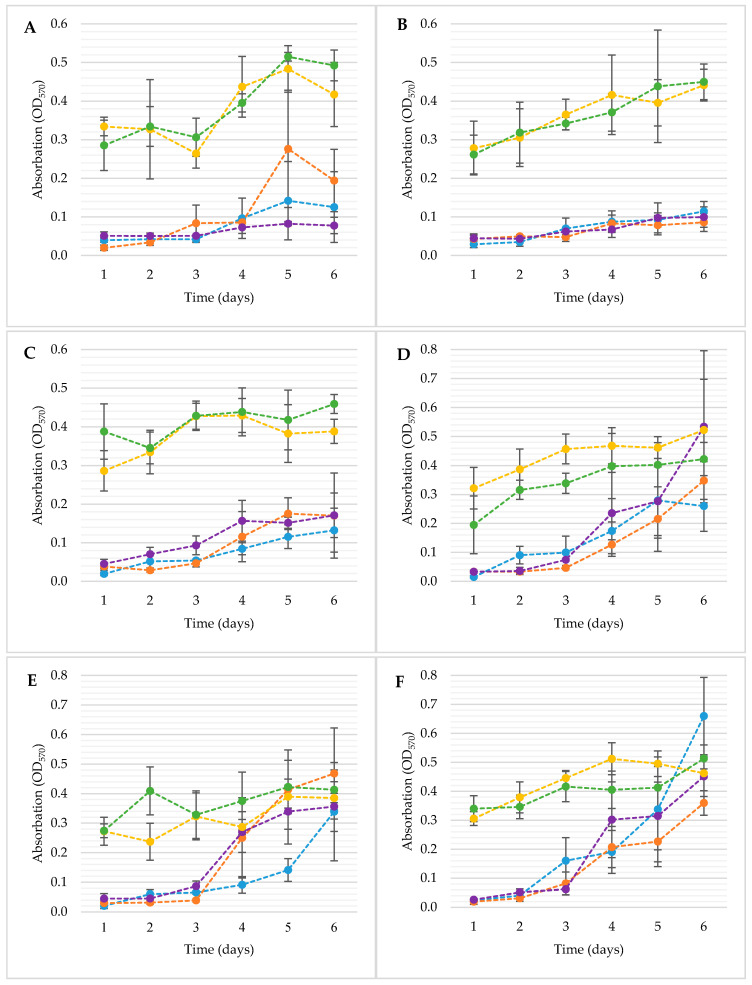
Formation of biofilms by *E. coli* ATCC 10536 (**A**); *E. coli* O157:H7 (**B**); *E.coli* ATCC 25922 (**C**); *S. enteritadis* ATCC 13076 (**D**); *S. enteritadis* ATCC 29631 (**E**); *S. hofit* IFM 2318 (**F**) on the surface of 24-well microplate for 6 days at 22 °C. C—control, not disinfected with any method; D1—disinfected with the 1st method; D2—disinfected with the 2nd method; DP1—disinfected with the 1st method + *L. plantarum* 299v; DP2—disinfected with the 2nd method + *L. plantarum* 299v; C —●; D1—●; D2—●; DP1—●; DP2—●; *n* = 4.

**Figure 5 foods-10-00542-f005:**
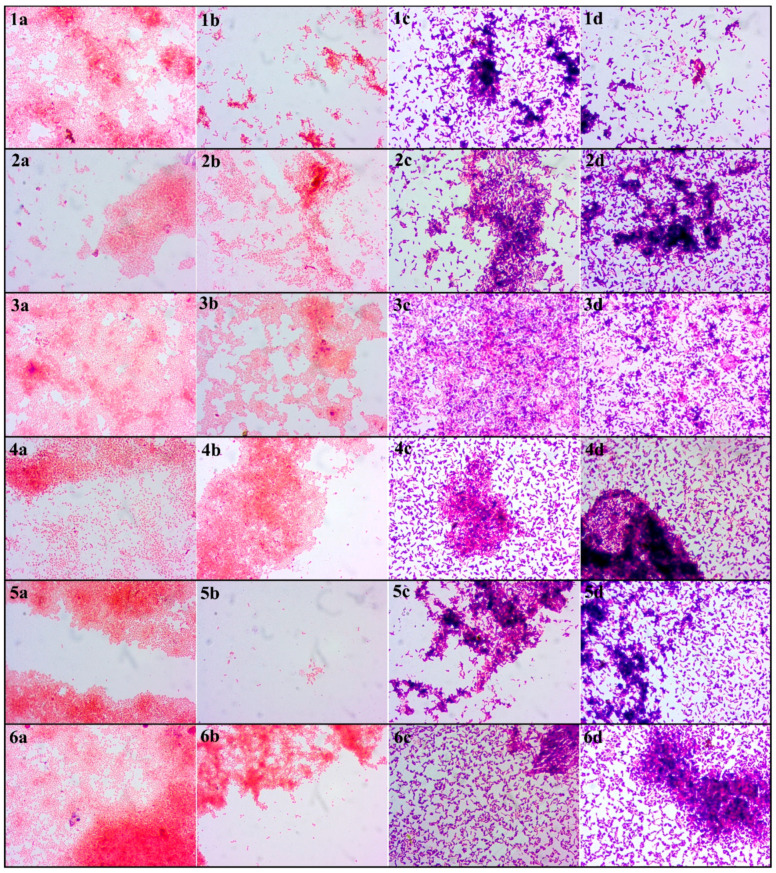
Light microscopy micrographs of biofilms formed during mung bean germination. Explanations: The numbers (**1**,**2**,**3**,**4**,**5**,**6**) refer to the strain. (**1**)—*E. coli* ATCC 10536; (**2**)—*E.coli* ATCC 25922; (**3**)—*E. coli* O157:H7; (**4**)—*S. enteritadis* ATCC 13076; (**5**)—*S. enteritadis* ATCC 29631; (**6**)—*S. hofit* IFM 2318. The letters (**a**,**b**,**c**,**d**) refer to the disinfection method. (**a**)—D1 disinfected with the 1st method; (**b**)—D2 disinfected with the 2nd method; (**c**)—DP1 disinfected with the 1st method + *L. plantarum* 299v; (**d**)—DP2 disinfected with the 2nd method + *L. plantarum* 299v; Bacteria were observed at 1000× magnification.

**Figure 6 foods-10-00542-f006:**
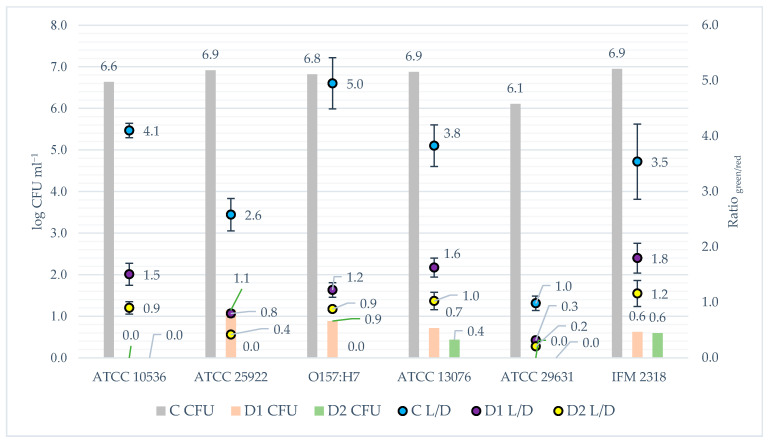
The survival of tested strains after disinfection procedure measured by fluorescence staining vs. CFU analysis. Explanations: L/D—LIVE/DEAD method = fluorescence staining; CFU—colony forming units method; bars refer to CFU results; dots refer to fluorescence staining analysis; C—control, not disinfected with any method; D1—disinfected with the 1st method; D2—disinfected with the 2nd method; DP1—disinfected with the 1st method + *L. plantarum* 299v; DP2—disinfected with the 2nd method + *L. plantarum* 299v.

**Table 1 foods-10-00542-t001:** The average number of *Escherichia coli* and *Lactobacillus plantarum* 299v on mung beans seeds sanitised by different methods and sprouted [log CFU g^−1^].

Sanitation Method ^1^	Time [Days]	*E. coli* ATCC 10536	*L. plantarum* 299v ^3^	*E. coli* ATCC 25922	*L. plantarum* 299v ^4^	*E. coli* O157:H7	*L. plantarum* 299v ^5^
C	0	6.63 ± 0.37 ^a^	n/a ^2^	6.92 ± 0.49 ^a^	n/a	6.82 ± 0.30 ^a^	n/a
	2	8.54 ± 0.13 ^b^	n/a	8.04 ± 0.17 ^a^	n/a	7.99 ± 0.32 ^abc^	n/a
	4	8.46 ± 0.09 ^b^	n/a	8.51 ± 0.35 ^a^	n/a	8.12 ± 0.12 ^bc^	n/a
	6	8.00 ± 0.07 ^b^	n/a	8.39 ± 0.06 ^a^	n/a	7.98 ± 0.18 ^abc^	n/a
D1	0	0.00 ± 0.00 _a_	n/a	1.72 ± 0.45 ^a^	n/a	0.88 ± 0.83 ^a^	n/a
	2	5.67 ± 0.01 ^b^	n/a	7.64 ± 0.08 ^b^	n/a	7.44 ± 0.20 ^b^	n/a
	4	7.81 ± 0.51 ^cd^	n/a	8.49 ± 0.04 ^b^	n/a	7.58 ± 0.07 ^b^	n/a
	6	7.24 ± 0.06 ^d^	n/a	8.68 ± 0.17 ^b^	n/a	7.73 ± 0.05 ^b^	n/a
D2	0	0.00 ± 0.00 ^a^	n/a	0.00 ± 0.00 ^a^	n/a	0.00 ± 0.00 ^a^	n/a
	2	0.00 ± 0.00 ^a^	n/a	3.59 ± 0.13 ^b^	n/a	5.48 ± 0.01 ^b^	n/a
	4	6.02 ± 0.06 ^b^	n/a	8.55 ± 0.04 ^c^	n/a	7.34 ± 0.09 ^c^	n/a
	6	8.34 ± 0.08 ^c^	n/a	8.54 ± 0.09 ^c^	n/a	7.99 ± 0.16 ^c^	n/a
DP1	0	0.00 ± 0.00 ^a^	n/a	1.15 ± 1.04 ^a^	n/a	0.88 ± 0.83 ^a^	n/a
	1	3.93 ± 0.54 ^b^	6.35 ± 0.32 ^a^	3.72 ± 0.65 ^b^	6.56 ± 0.10 ^a^	6.10 ± 0.24 ^b^	6.65 ± 0.16 ^a^
	2	6.65 ± 0.10 ^c^	8.37 ± 0.04 ^b^	8.17 ± 0.15 ^c^	7.94 ± 0.24 ^b^	7.57 ± 0.08 ^c^	8.34 ± 0.06 ^b^
	4	7.60 ± 0.03 ^d^	8.70 ± 0.02 ^b^	8.26 ± 0.09 ^c^	8.58 ± 0.38 ^b^	7.46 ± 0.05 ^c^	8.81 ± 0.24 ^bc^
	6	7.69 ± 0.10 ^d^	8.61 ± 0.06 ^b^	8.48 ± 0.10 ^c^	8.42 ± 0.10 ^b^	7.35 ± 0.03 ^c^	7.57 ± 0.88 ^a^
DP2	0	0.00 ± 0.00 ^a^	n/a	0.00 ± 0.00 ^a^	n/a	0.00 ± 0.00 ^a^	n/a
	1	0.00 ± 0.00 ^a^	6.74 ± 0.16 ^a^	0.00 ± 0.00 ^a^	6.57 ± 0.34 ^a^	2.11 ± 1.84 ^b^	6.52 ± 0.12 ^a^
	2	6.51 ± 0.04 ^b^	8.48 ± 0.47 _b_	2.18 *^a^	7.62 ± 0.09 ^b^	6.13 ± 0.05 ^c^	8.04 ± 0.03 ^b^
	4	5.49 ± 0.08 ^b^	8.43 ± 0.34 ^b^	5.55 ± 0.11 ^b^	8.23 ± 0.11 ^bc^	3.77 ± 0.30 ^b^	8.23 ± 0.16 ^b^
	6	7.12 ± 0.51 ^c^	7.67 ± 0.05 ^c^	7.73 ± 0.81 ^c^	8.49 ± 0.20 ^c^	6.81 ± 1.06 ^c^	8.40 ± 0.10 ^b^

Explanations: ^1^—C—control, not disinfected with any method; D1—disinfected with the 1st method; D2—disinfected with the 2nd method; DP1—disinfected with the 1st method + *L. plantarum* 299v; DP2—disinfected with the 2nd method + *L. plantarum* 299v; ^2^—not applicable; ^3^—count of lactobacillus plantarum used for the *E. coli* ATCC 10536; ^4^—count of *Lactobacillus plantarum* used for the *E. coli* ATCC 25922; ^5^—count of *Lactobacillus plantarum* used for the *E. coli* O157:H7; *—single replicate result, the other two assay results were < 1 log CFU g^−1^; ^a,b,c,d^ are significantly different (*p* < 0.05), significant differences were marked between individual days within one sanitation method, *n* = 3.

**Table 2 foods-10-00542-t002:** The average number of *Salmonella* and *Lactobacillus plantarum* 299v on mung beans seeds sanitised by different methods and sprouted [log CFU g^−1^].

Sanitation Method ^1^	Time [Days]	*S. enteritidis* ATCC 13076	*L. plantarum* 299v ^3^	*S. enteritidis* ATCC 29631	*L. plantarum* 299v ^4^	*S. hofit* IFM 2318	*L. plantarum* 299v ^5^
C	0	6.88 ± 0.34 ^a^	n/a ^2^	6.11 ± 0.11 ^ac^	n/a	6.95 ± 0.34 ^ac^	n/a
	2	8.55 ± 0.02 ^bc^	n/a	8.35 ± 0.00 ^bcd^	n/a	8.58 ± 0.08 ^bcd^	n/a
	4	7.72 ± 0.56 ^ac^	n/a	8.32 ± 0.14 ^abcd^	n/a	8.46 ± 0.05 ^abcd^	n/a
	6	8.84 ± 0.22 ^bc^	n/a	8.56 ± 0.36 ^bcd^	n/a	8.81 ± 0.18 ^bcd^	n/a
D1	0	2.14 *^a^	n/a	0.00 ± 0.00 ^a^	n/a	1.87 *^a^	n/a
	2	7.40 ± 0.14 ^b^	n/a	4.12 ± 0.08 ^bc^	n/a	7.42 ± 0.06 ^b^	n/a
	4	8.20 ± 0.32 ^b^	n/a	3.18 ± 0.00 ^c^	n/a	8.81 ± 0.06 ^b^	n/a
	6	8.19 ± 0.02 ^b^	n/a	2.91 ± 0.37 ^c^	n/a	8.30 ± 0.22 ^b^	n/a
D2	0	1.30 *^a^	n/a	0.00 ± 0.00 ^a^	n/a	1.78 *^a^	n/a
	2	2.85 ± 0.19 ^b^	n/a	0.00 ± 0.00 ^a^	n/a	7.44 ± 0.04 ^b^	n/a
	4	8.29 ± 0.06 ^c^	n/a	0.00 ± 0.00 ^a^	n/a	7.85 ± 0.19 ^b^	n/a
	6	8.55 ± 0.16 ^c^	n/a	0.00 ± 0.00 ^a^	n/a	8.14 ± 0.06 ^b^	n/a
DP1	0	2.15 *^a^	n/a	0.00 ± 0.00 ^a^	n/a	1.88 *^a^	n/a
	1	5.83 ± 0.05 ^b^	6.76 ± 0.09 ^a^	0.00 ± 0.00 ^a^	6.37 ± 0.16 ^a^	2.90 ± 0.05 ^b^	6.57 ± 0.20 ^a^
	2	6.46 ± 0.13 ^c^	7.61 ± 0.04 ^b^	2.62 ± 0.54 ^bc^	7.98 ± 0.04 ^b^	8.28 ± 0.05 ^c^	8.13 ± 0.05 ^b^
	4	8.27 ± 0.14 ^d^	8.67 ± 0.44 ^c^	3.67 ± 0.17 ^bc^	8.48 ± 0.23 ^c^	8.62 ± 0.07 ^c^	8.58 ± 0.33 ^bc^
	6	8.66 ± 0.09 ^d^	8.73 ± 0.06 ^c^	2.74 *^ab^	8.53 ± 0.18 ^c^	8.34 ± 0.17 ^c^	8.64 ± 0.03 ^c^
DP2	0	1.30 *^a^	n/a	0.00 ± 0.00 ^a^	n/a	1.78 *^a^	n/a
	1	2.65 *^a^	6.37 ± 0.17 ^a^	2.18 *^a^	6.52 ± 0.21 ^a^	0.00 ± 0.00 ^a^	6.49 ± 0.02 ^a^
	2	7.83 ± 0.11 ^b^	8.41 ± 0.22 ^b^	0.00 ± 0.00 ^a^	7.98 ± 0.09 ^b^	5.18 ± 0.00 ^b^	8.17 ± 0.04 ^b^
	4	8.18 ± 0.09 ^b^	8.19 ± 0.25 ^b^	0.00 ± 0.00 ^a^	8.40 ± 0.09 ^b^	7.35 ± 0.03 ^c^	8.37 ± 0.16 ^b^
	6	7.97 ± 0.24 ^b^	8.38 ± 0.44 ^b^	0.00 ± 0.00 ^a^	8.25 ± 0.07 ^b^	8.21 ± 0.16 ^c^	8.50 ± 0.05 ^b^

Explanations: ^1^—C—control, not disinfected with any method; D1—disinfected with the 1st method; D2—disinfected with the 2nd method; DP1—disinfected with the 1st method + *L. plantarum* 299v; DP2—disinfected with the 2nd method + *L. plantarum* 299v; ^2^—not applicable ^3^—count of *Lactobacillus plantarum* used for the *S. enteritidis* ATCC 13076; ^4^—count of *Lactobacillus plantarum* used for the *S. enteritidis* ATCC 29631; ^5^—count of *Lactobacillus plantarum* used for the *S. Hofit* IFM 2318; *—single replicate result, the other two assay results were < 1 log CFU g^−1^; ^a,b,c,d^ are significantly different (*p* < 0.05), significant differences were marked between individual days within one sanitation method; *n* = 3.

**Table 3 foods-10-00542-t003:** Fluorescence intensity of gram-negative bacteria in analysed samples based on Ratio green/red.

Sanitation Method	Time [Days]	*E. coli* ATCC 10536	*E. coli* ATCC 25922	*E. coli* O157:H7	*S. enteritidis* ATCC 13076	*S. enteritidis* ATCC 29631	*S. hofit* IFM 2318
CP	1	2.18 ± 0.33 ^a^	3.45 ± 0.31 ^a^	3.49 ± 0.26 ^a^	1.94 ± 0.16 ^a^	1.53 ± 0.19 ^a^	3.67 ± 0.11 ^a^
	2	1.40 ± 0.13 ^b^	1.47 ± 0.22 ^bd^	3.13 ± 0.11 ^ac^	2.65 ± 0.28 ^b^	2.00 ± 0.36 ^a^	2.93 ± 0.19 ^bcd^
	4	2.99 ± 0.20 ^c^	1.73 ± 0.12 ^bde^	2.44 ± 0.06 ^bc^	2.53 ± 0.09 ^b^	1.87 ± 0.08 ^a^	2.59 ± 0.17 ^bc^
	6	3.75 ± 0.17 ^d^	2.10 ± 0.44 ^cde^	2.92 ± 0.22 ^bc^	2.91 ± 0.22 ^b^	2.67 ± 0.13 ^b^	3.24 ± 0.05 ^bd^
DP1	1	1.45 ± 0.04 ^ac^	1.29 ± 0.09 ^a^	1.83 ± 0.17 ^a^	1.45 ± 0.12 ^a^	1.26 ± 0.19 ^ac^	1.72 ± 0.08 ^a^
	2	1.22 ± 0.04 ^a^	1.04 ± 0.04 ^a^	1.82 ± 0.28 ^a^	1.43 ± 0.20 ^a^	1.13 ± 0.13 ^ac^	1.45 ± 0.19 ^a^
	4	1.83 ± 0.06 ^ac^	1.43 ± 0.23 ^a^	1.84 ± 0.24 ^a^	2.00 ± 0.07 ^b^	1.52 ± 0.27 ^acd^	1.84 ± 0.14 ^a^
	6	2.32 ± 0.26 ^b^	2.24 ± 0.14 ^b^	2.26 ± 0.25 ^b^	2.37 ± 0.12 ^b^	1.94 ± 0.04 ^bcd^	2.33 ± 0.14 ^b^
DP2	1	1.49 ± 0.11 ^acd^	1.42 ± 0.14 ^a^	1.42 ± 0.11 ^a^	1.44 ± 0.15 ^a^	1.22 ± 0.07 ^a^	1.44 ± 0.11 ^a^
	2	1.35 ± 0.04 ^ac^	1.08 ± 0.02 ^a^	1.18 ± 0.11 ^a^	1.25 ± 0.04 ^a^	1.41 ± 0.12 ^a^	1.41 ± 0.11 ^a^
	4	1.94 ± 0.22 ^ad^	1.44 ± 0.09 ^a^	2.08 ± 0.10 ^b^	2.09 ± 0.04 ^b^	1.62 ± 0.20 ^a^	2.09 ± 0.22 ^b^
	6	2.75 ± 0.29 ^b^	2.56 ± 0.03 ^b^	2.87 ± 0.18 ^c^	2.28 ± 0.25 ^b^	2.16 ± 0.22 ^b^	2.50 ± 0.22 ^c^

Explanations: CP—control, not disinfected with any method + *L. plantarum* 299v; DP1—disinfected with the 1st method + *L. plantarum* 299v; DP2—disinfected with the 2nd method + *L. plantarum* 299v; ^a,b,c,d,e^ are significantly different (*p* < 0.05), significant differences were marked between individual days within one sanitation method; *n* = 4.

## Data Availability

No new data were created or analysed in this study. Data sharing is not applicable to this article.

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
