# Peer review of "Analysis of Biofilm Formation on the Surface of Organic Mung Bean Seeds, Sprouts and in the Germination Environment"

_foods, 2021, doi:10.3390/foods10030542_

Round 1
Reviewer 1 Report
The article "Analysis of biofilm formation on the surface of organic mung bean seeds, sprouts and in the germination environment" will be of interest to many readers. The authors employed several methods to investigate biofilm formation in germinated organic mung bean seeds.
However, the following need to be revised:
L22: ...will let to the complete elimination.. do you mean: "will lead to the complete..."?
L26-27: ..is gaining more popularity among consumers.
L34: fungi (plural) instead of fungus (singular)
L42-44: revise for clarity
L48-40: revise for clarity
On page 3: in many instances, the authors write "storaged" instead of "stored"
L114: It is better to "For the bean seed inoculation, L. plantarum 299v was used as a biological sanitation factor,...
L123: at what temperature were they allowed to dry?
The results are well-presented with figures and tables. The discussion can made more robust with recent citations on the topic.
L489: Perhaps: "The growth dynamic of bacteria presented in this experiment..."
L507: ...more or less capable of forming biofilms.
L529-533: Revise for clarity and highlight the inactivating effect of L. plantarum 229v in this study to other similar studies.
L562-564: Revise for clarity.
Author Response
The answers are in the attached file.

Reviewer 2 Report
- The exact source for E. coli O157:H7 is not provided.
- In Line 194, the culture temperature of the two strains was cultured at 22 degrees for 24 hours. What is the reason? The optimum temperature for the two bacteria is about 35 degrees, but I am curious about the reason for the treatment at a low temperature. Also, what is the initial microbial treatment concentration used in the experiment?
- If the bottom of the well is scraped in Line 200, the film will be fragmented or filtered. Will the bio film be confirmed later?
- The picture shown in fig.5 appears similar even when the concentration of bacteria is high. Is it possible to check the composition of the bio-film by other methods?
- In Tables 1 and 2, E.coli and Salmonella species all appear more than 8 log and the concentration of lactic acid bacteria is also more than 8 log. The picture looks awkward if there is a color change as in 5.
Author Response
The answers are in the attached file.

Reviewer 3 Report
Abstract
It is always advised to define acronyms when we need to use them later. However, there are acronyms that are not defined (D2, DP2).
The introduction is very well written. I was wondering why the authors chose mung bean as material. Could you please define it?
Figure 1. Figures should be easy to understand as standalone objects. Please define all the used acronyms in the legend. Additionally, add some explanation to the legend.
The description of the methods is clear and concise.
Statistical analysis should be corrected. Did the authors use MANOA or ANOVA? What correlation was used? Did the authors test normality?
Results.
When presenting the results of statistical tests, the proper presentation method should be used. If the authors, used ANOVA, then the following paper might be of help: http://ich.vscht.cz/~svozil/lectures/vscht/2015_2016/sad/APA_style2.pdf
Please correct this for all cases when statistical tests are reported.
Figure 4 is well-edited, however, I would put the legend into the title of the figure with colored points. Therefore, the plots would look a bit better.
The paper is well-written and presents interesting results not only for academics but for industry professionals as well. As the authors measured, analyzed, and presented multiple data sets, it came to my mind if they would want to compare the different treatments using multicriteria decision-making methods. For your information, I link here a recent paper introducing a very easy but robust method: https://doi.org/10.1016/j.foodchem.2020.128617
Author Response
The answers are in the attached file.
